# Excess Risk Bounds for the Bayes Risk using Variational Inference in Latent Gaussian Models

**Rishit Sheth** and **Roni Khardon**
Department of Computer Science, Tufts University
Medford, MA, 02155, USA
`rishit.sheth@tufts.edu` | `roni@cs.tufts.edu`

## Abstract

Bayesian models are established as one of the main successful paradigms for complex problems in machine learning. To handle intractable inference, research in this area has developed new approximation methods that are fast and effective. However, theoretical analysis of the performance of such approximations is not well developed. The paper furthers such analysis by providing bounds on the excess risk of variational inference algorithms and related regularized loss minimization algorithms for a large class of latent variable models with Gaussian latent variables. We strengthen previous results for variational algorithms by showing that they are competitive with any point-estimate predictor. Unlike previous work, we provide bounds on the risk of the Bayesian predictor and not just the risk of the Gibbs predictor for the same approximate posterior. The bounds are applied in complex models including sparse Gaussian processes and correlated topic models. Theoretical results are complemented by identifying novel approximations to the Bayesian objective that attempt to minimize the risk directly. An empirical evaluation compares the variational and new algorithms shedding further light on their performance.

## 1 Introduction

Bayesian models are established as one of the main successful paradigms for complex problems in machine learning. Since inference in complex models is intractable, research in this area is devoted to developing new approximation methods that are fast and effective (Laplace/Taylor approximation, variational approximation, expectation propagation, MCMC, etc.), i.e., these can be seen as algorithmic contributions. Much less is known about theoretical guarantees on the loss incurred by such approximations, either when the Bayesian model is correct or under model misspecification.

Several authors provide risk bounds for the Bayesian predictor (that aggregates predictions over its posterior and then predicts), e.g., see [15, 6, 12]. However, the analysis is specialized to certain classification or regression settings, and the results have not been shown to be applicable to complex Bayesian models and algorithms like the ones studied in this paper.

In recent work, [7] and [1] identified strong connections between *variational inference* [10] and PAC-Bayes bounds [14] and have provided oracle inequalities for variational inference. As we show in Section 3, similar results that are stronger in some aspects can be obtained by viewing variational inference as performing regularized loss minimization. These results are an exciting first step, but they are limited in two aspects. First, they hold for the Gibbs predictor (that samples a hypothesis and uses it to predict) and not the Bayesian predictor and, second, they are only meaningful against "weak" competitors. For example, the bounds go to infinity if the competitor is a point estimate with zero variance. In addition, these results do not explicitly address hierarchical Bayesian models

where further development is needed to distinguish among different variational approximations in the literature. Another important result by [11] provides relative loss bounds for generalized linear models (GLM). These bounds can be translated to risk bounds and they hold against point estimates. However, they are limited to the prediction of the true Bayesian posterior which is hard to compute.

In this paper we strengthen these theoretical results and, motivated by these, make additional algorithmic and empirical contributions. In particular, we focus on latent Gaussian models (LGM) whose latent variables are normally distributed. We extend the technique of [11] to derive agnostic bounds for the excess risk of an approximate Bayesian predictor against any point estimate competitor. We then apply these results to several models with two levels of latent variables, including generalized linear models (GLM), sparse Gaussian processes (sGP) [17, 26] and correlated topic models (CTM) [3] providing high probability bounds for risk. For CTM our results apply precisely to the variational algorithm and for GLM and sGP they apply for a variant with a smoothed loss function.

Our results improve over [7, 1] by strengthening the bounds, showing that they can be applied directly to the variational algorithm, and showing that they apply to the Bayesian predictor. On the other hand they improve over [11] in analyzing the approximate inference algorithms and in showing how to apply the bounds to a larger class of models.

Finally, viewing approximate inference as regularized loss minimization, our exploration of the hierarchical models shows that there is a mismatch between the objective being optimized by algorithms such as variational inference and the loss that defines our performance criterion. We identify three possible objectives corresponding respectively to a "simple variational approximation", the "collapsed variational approximation", and to a new algorithm performing direct regularized loss minimization instead of optimizing the variational objective. We explore these ideas empirically in CTM. Experimental results confirm that each variant is the "best" for optimizing its own implicit objective, and therefore direct loss minimization, for which we do not yet have a theoretical analysis, might be the algorithm of choice. However, they also show that the collapsed approximation comes close to direct loss minimization. The concluding section of the paper further discusses the results.

## 2 Preliminaries

### 2.1 Learning Model, Hypotheses and Risk

We consider the standard PAC setting where $n$ samples are drawn i.i.d. according to an unknown joint distribution $D$ over the sample space $z$. This captures the supervised case where $z = (x, y)$ and the goal is to predict $y|x$. In the unsupervised case, $z = y$ and we are simply modeling the distribution. To treat both cases together we always include $x$ in the notation but fix it to a dummy value in the unsupervised case.

A learning algorithm outputs a hypothesis $h$ which induces a distribution $p_h(y|x)$. One would normally use this predictive distribution and an application-specific loss to pick the prediction. Following previous work, we primarily focus on log loss, i.e., the loss of $h$ on example $(x_*, y_*)$ is $\ell(h, (x_*, y_*)) = -\log p_h(y_*|x_*)$. In cases where this loss is not bounded, a smoothed and bounded variant of the log loss can be defined as $\tilde{\ell}(h, (y_*, x_*)) = -\log\big((1-\alpha)p_h(y|x) + \alpha\big)$, where $0 < \alpha < 1$. We state our results w.r.t. log loss, and demonstrate, by example, how the smoothed log loss can be used. Later, we briefly discuss how our results hold more generally for losses that are convex in $p$.

We start by considering one-level (1L) latent variable models given by $p(w)p(y|w, x)$ where $p(y|w, x) = \prod_i p(y_i|w, x_i)$. For example, in Bayesian logistic regression, $w$ is the hidden weight vector, the prior $p(w)$ is given by a Normal distribution $\mathcal{N}(w|\mu, \Sigma)$ and the likelihood term is $p(y|w, x) = \sigma(yw^T x)$ where $\sigma()$ is the sigmoid function. A hypothesis $h$ represents a distribution $q(w)$ over $w$, where point estimates for $w$ are modeled as delta functions. Regardless of how $h$ is computed, the *Bayesian predictor* calculates a predictive distribution $p_h(y|x) = \mathrm{E}_{q(w)}[p(y|w, x)]$ and accordingly its risk is defined as $r_{\mathrm{Bay}}(q(w)) = \mathrm{E}_{(x,y)\sim D}[-\log p_h(y|x)] = \mathrm{E}_{(x,y)\sim D}[-\log \mathrm{E}_{q(w)}[p(y|w, x)]]$.

Following previous work we also analyze the average risk of the *Gibbs predictor* which draws a random $w$ from $q(w)$ and predicts using $p(y|w, x)$. Although the Gibbs predictor is not an optimal strategy, its analysis has been found useful in previous work and it serves as an intermediate step

in our results. Assuming the draw of $w$ is done independently for each $x$ we get: $r_{\text{Gib}}(q(w)) = \text{E}_{(x,y)\sim D}[\text{E}_{q(w)}[-\log p(y|w, x)]]$. Previous work has defined the Gibbs risk with expectations in reversed order. That is, the algorithm draws a single $w$ and uses it for prediction on all examples. We find the one given here more natural. Some of our results require the two definitions to be equivalent, i.e., the conditions for Fubini's theorem must hold. We make this explicit in

**Assumption 1.** $\text{E}_{(x,y)\sim D}[\text{E}_{q(w)}[-\log p(y|w, x)]] = \text{E}_{q(w)}[\text{E}_{(x,y)\sim D}[-\log p(y|w, x)]]$.

This is a relatively mild assumption. It clearly holds when $y$ takes discrete values, where $p(y|x, w) \leq 1$ implies that the log loss is positive and Fubini's theorem applies. In the case of continuous $y$, upper bounded likelihood functions imply that a translation of the loss function satisfies the condition of Fubini's theorem. For example, if $p(y|x, w) = \mathcal{N}(y|f(w, x), \sigma^2)$ where $\sigma^2$ is a hyperparameter, then $\log p(y|x, w) \leq B = -\log(\sqrt{2\pi}) - \log(\sigma^2)$. Therefore, $-\log p(y|x, w) + B \geq 0$ so that if we redefine[1] the loss by adding the constant $B$, then the loss is positive and Fubini's theorem applies. More generally, we might need to enforce constraints on $D$, $q(w)$, and/or $p(y|x, w)$.

## 2.2 Variational Learners for Latent Variable Models

Approximate inference generally limits $q(w)$ to some fixed family of distributions $Q$ (e.g. the family of normal distributions, or the family of products of independent components in the mean-field approximation). Given a dataset $S = \{(x_i, y_i)\}_{i=1}^n$, we define the following general problem,

$$q^\star = \arg\min_{q \in Q} \left\{ \frac{1}{\eta} \text{KL}\left(q(w)\|p(w)\right) + L(w, S) \right\}, \tag{1}$$

where KL denotes Kullback-Leibler divergence. Standard variational inference uses $\eta = 1$ and $L(w, S) = -\sum_i \text{E}_{q(w)}[\log p(y_i|w, x_i)]$, and it is well known that (1) is the optimization of a lower bound on $p(y)$. If $-\log p(y_i|w, x_i)$ is replaced with a general loss function, then (1) may no longer correspond to a lower bound on $p(y)$. In any case, the output of (1), denoted by $q^\star_{\text{Gib}}$, is achieved via regularized cumulative-loss minimization (RCLM) which optimizes a sum of training set error and a regularization function. In particular, $q^\star_{\text{Gib}}$ uses a KL regularizer and optimizes the Gibbs risk $r_{\text{Gib}}$ in contrast to the Bayes risk $r_{\text{Bay}}$. This motivates some of the analysis in the paper.

Many interesting Bayesian models have two levels (2L) of latent variables given by $p(w)p(f|w, x)\prod_i p(y_i|f_i)$ where both $w$ and $f$ are latent. Of course one can treat $(w, f)$ as one set of parameters and apply the one-level model, but this does not capture the hierarchical structure of the model. The standard approach in the literature infers a posterior on $w$ via a variational distribution $q(w)q(f|w)$, and assumes that $q(w)$ is sufficient for predicting $p(y_*|x_*)$. We refer to this structural assumption, i.e., $p(f_*, f|w, x, x_*) = p(f_*|w, x_*)p(f|w, x)$, as *Conditional Independence*. It holds in models where an additional factorization $p(f|w, x) = \prod_i p(f_i|w, x_i)$ holds, e.g., in GLM, CTM. In the case of sparse Gaussian processes (sGP), Conditional Independence does not hold, but it is required in order to reduce the cubic complexity of the algorithm, and it has been used in all prior work on sGP. Assuming Conditional Independence, the definition of risk extends naturally from the one-level model by writing $p(y|w, x) = \text{E}_{p(f|w, x)}[p(y|f)]$ to get:

$$r_{\text{2Bay}}(q(w)) = \underset{(x,y)\sim D}{\text{E}}[-\log \underset{q(w)}{\text{E}}[\underset{p(f|w, x)}{\text{E}}[p(y|f)]]], \tag{2}$$

$$r_{\text{2Gib}}(q(w)) = \underset{(x,y)\sim D}{\text{E}}[\underset{q(w)}{\text{E}}[-\log \underset{p(f|w, x)}{\text{E}}[p(y|f)]]]. \tag{3}$$

Even though Conditional Independence is used in prediction, the learning algorithm must decide how to treat $q(f|w)$ during the optimization of $q(w)$. The mean field approximation uses $q(w)q(f)$ in the optimization. We analyze two alternatives that have been used in previous work. The approximation $q(f|w) = p(f|w)$, used in sparse GP [26, 8, 23], is described by (1) with $L(w, S) = -\sum_i \text{E}_{q(w)}[\text{E}_{p(f_i|w, x_i)}[\log p(y_i|f_i)]]$. We denote this by $q^\star_{\text{2A}}$ and observe it is the RCLM solution for the risk defined as

$$r_{\text{2A}}(q(w)) = \underset{(x,y)\sim D}{\text{E}}[\underset{q(w)}{\text{E}}[\underset{p(f|w, x)}{\text{E}}[-\log p(y|f)]]]. \tag{4}$$

As shown by [25, 9, 22], alternatively, for each $w$, we can pick the optimal $q(f|w) = p(f|w, S)$. Following [25] we call this a *collapsed approximation*. This leads to (1) with $L(w, S) = -\operatorname{E}_{q(w)}[\log \operatorname{E}_{p(f|w,x)}[\prod_i p(y_i|f_i)]]$ and is denoted by $q^\star_{2\mathrm{Bj}}$ (joint expectation). For models where $p(f|w) = \prod_i p(f_i|w)$, this simplifies to $L(w, S) = -\sum_i \operatorname{E}_{q(w)}[\log \operatorname{E}_{p(f_i|w,x_i)}[p(y_i|f_i)]]$, and we denote the algorithm by $q^\star_{2\mathrm{Bi}}$ (independent expectation). Note that $q^\star_{2\mathrm{Bi}}$ performs RCLM for the risk given by $r_{2\mathrm{Gib}}$ even if the factorization does not hold.

Finally, viewing approximate inference as performing RCLM, we observe a discrepancy between our definition of risk in (2) and the loss function being optimized by existing algorithms, e.g., variational inference. This perspective suggests direct loss minimization described by the alternative $L(w, S) = -\sum_i \log \operatorname{E}_{q(w)}[\operatorname{E}_{p(f_i|w,x_i)}[p(y_i|f_i)]]$ in (1) and which we denote $q^\star_{2\mathrm{D}}$. In this case, $q^\star_{2\mathrm{D}}$ is a "posterior" but one for which we do not have a Bayesian interpretation.

Given the discussion so far, we can hope to get some analysis for regularized loss minimization where each of the algorithms implicitly optimizes a different definition of risk. Our goal is to identify good algorithms for which we can bound the definition of risk we care about, $r_{2\mathrm{Bay}}$, as defined in (2).

## 3 RCLM

Regularized loss minimization has been analyzed for general hypothesis spaces and losses. For hypothesis space $H$ and hypothesis $h \in H$ we have loss function $\ell(h, (x, y))$, and associated risk $r(h) = \operatorname{E}_{(x,y) \sim D}[\ell(h, (x, y))]$. Now, given a regularizer $R : H \to 0 \cup \mathbb{R}_+$, a non-negative scalar $\eta$, and sample $S$, regularized cumulative loss minimization is defined as

$$\mathrm{RCLM}(H, \ell, R, \eta, S) = \underset{h \in H}{\arg\min} \left( \frac{1}{\eta} R(h) + \sum_i \ell(h, (x_i, y_i)) \right). \tag{5}$$

**Theorem 1** ([20][2] ). *Assume that the regularizer $R(h)$ is $\sigma$-strong-convex in $h$ and the loss $\ell(h, (x, y))$ is $\rho$-Lipschitz and convex in $h$, and let $h^\star(S) = RCLM(H, \ell, R, \eta, S)$. Then, for all $h \in H$, $\operatorname{E}_{S \sim D^n}[r(h^\star(S))] \leq r(h) + \frac{1}{\eta n} R(h) + \frac{4\rho^2 \eta}{\sigma}$.*

The theorem bounds the expectation of the risk. Using Markov's inequality we can get a high probability bound: with probability $\geq 1 - \delta$, $r(h^\star(S)) \leq r(h) + \frac{1}{\delta}(\frac{1}{\eta n} R(h) + \frac{4\rho^2 \eta}{\sigma})$. Tighter dependence on $\delta$ can be achieved for bounded losses using standard techniques. To simplify the presentation we keep the expectation version throughout the paper.

For this paper we specialize RCLM for Bayesian algorithms, that is, $H$ corresponds to the parameter space for a parameterized family of (possibly degenerate) distributions, denoted $Q$, where $q \in Q$ is a distribution over a base parameter space $w$.

We have already noted above that $q^\star_{\mathrm{Gib}}(w)$, $q^\star_{2\mathrm{Bi}}(w)$ and $q^\star_{2\mathrm{D}}(w)$ are RCLM algorithms. We can therefore get immediate corollaries for the corresponding risks (see supplementary material). Such results are already useful, but the convexity and $\rho$-Lipschitz conditions are not always easy to analyze or guarantee. We next show how to use recent ideas from PAC-Bayes analysis to derive a similar result for Gibbs risk with less strong requirements. We first develop the result for the one-level model. Toward this, define the loss and risk for individual base parameters as $\ell_W(w, (x, y))$, and $r_W(w) = \operatorname{E}_D[\ell_W(w, (x, y))]$, and the empirical estimate $\hat{r}_W(w, S) = \frac{1}{n} \sum_i \ell_W(w, (x_i, y_i))$. Following [7], let $\Psi(\lambda, n) = \log \operatorname{E}_{S \sim D^n}[\operatorname{E}_{p(w)}[e^{\lambda(r_W(w) - \hat{r}_W(w,S))}]]$ where $\lambda$ is an additional parameter. Combining arguments from [20] with the use of the compression lemma [2] as in [7] we can derive the following bound (proof in supplementary material):

**Theorem 2.** *For all $q \in Q$, $\operatorname{E}_{S \sim D^n}[r_{Gib}(q^\star_{Gib}(w))] \leq r_{Gib}(q) + \frac{1}{\eta n} KL\left(q \| p\right) + \frac{1}{\lambda} \max_{q \in Q} KL\left(q \| p\right) + \frac{1}{\lambda} \Psi(\lambda, n)$.*

The theorem applies to the two-level model by writing $p(y|w) = \operatorname{E}_{p(f|w)}[p(y|f)]$. This yields

**Corollary 3.** *For all $q \in Q$, $\operatorname{E}_{S \sim D^n}[r_{2Gib}(q^\star_{2Bi}(w))] \leq r_{2Gib}(q) + \frac{1}{\eta n} KL\left(q \| p\right) + \frac{1}{\lambda} \max_{q \in Q} KL\left(q \| p\right) + \frac{1}{\lambda} \Psi(\lambda, n)$.*

A similar result has already been derived by [1] without making the explicit connection to RCLM. However, the implied algorithm uses a "regularization factor" $\lambda$ which may not coincide with $\eta = 1$, whereas standard variational inference can be analyzed with Theorem 2 (or Corollary 3).

The work of [4, 7] showed how the $\Psi$ term can be bounded. Briefly, if $\ell_W(w, (x, y))$ is bounded in $[a, b]$, then $\Psi(\lambda, n) \leq \frac{\lambda^2 (b-a)^2}{2n}$; if $\ell_W(w, (x, y))$ is not bounded, but the random variable $r_W(w) - \ell_W(w, (x, y))$ is sub-Gaussian or sub-gamma, then $\Psi(\lambda, n)$ can be bounded with additional assumptions on the underlying distribution $D$. More details are in the supplementary material.

## 4 Concrete Bounds on Excess Risk in LGM

The LGM family is a special case of the two-level model where the prior $p(w)$ over the $M$-dimensional parameter $w$ is given by a Normal distribution. Following previous work we let $Q$ to be a family of Normal distributions. For the analysis we further restrict $Q$ by placing bounds on the mean and covariance as follows: $Q = \{\mathcal{N}(w|m, V) \text{ s.t. } \|m\|_2 \leq B_m, \lambda_{\min}(V) \geq \epsilon, \lambda_{\max}(V) \leq B_V\}$ for some $\epsilon > 0$. The KL divergence from $q(w) = \mathcal{N}(w|m, V)$ to $p(w) = \mathcal{N}(w|\mu, \Sigma)$ is given by $\text{KL}(q\|p) = \frac{1}{2}\left(\text{tr}(\Sigma^{-1}V) + (\mu - m)^T\Sigma^{-1}(\mu - m) + \log\frac{|\Sigma|}{|V|} - M\right)$.

### 4.1 General Bounds on Excess Risk in LGM Against Point Estimates

First, we note that $\text{KL}(q\|p)$ is bounded under a lower bound on the minimum eigenvalue of $V$ (proof in supplementary material follows from linear algebra identities):

**Lemma 4.** *Let* $B'_R = \frac{1}{2}\left(\frac{MB_V + \|\mu\|_2^2 + B_m^2}{\lambda_{min}(\Sigma)} + M\log(\lambda_{max}(\Sigma)) - M\right)$. *For* $q \in Q$,

$$KL(q\|p) \leq B_R = \frac{1}{2}\left(\frac{MB_V + \|\mu\|_2^2 + B_m^2}{\lambda_{min}(\Sigma)} + M\log\left(\frac{\lambda_{max}(\Sigma)}{\epsilon}\right) - M\right) = B'_R - \frac{1}{2}M\log\epsilon. \quad (6)$$

The risk bounds of the previous section do not allow for point estimate competitors because the KL portion is not bounded. We next generalize a technique from [11] showing that adding a little variance to a point estimate does not hurt too much. This allows us to derive the promised bounds. In the following, $\epsilon > 0$ is a constant whose value is determined in the proof. For any $\hat{w}$, we consider the $\epsilon$-inflated distribution $q(w) = \mathcal{N}(w|\hat{w}, \epsilon I)$ and calculate the distribution's Gibbs risk w.r.t. a generic loss. Specifically, we consider the (1L or 2L) Gibbs risk $r(q) = \text{E}_{(x,y)\sim D}[\text{E}_{q(w)}[\ell(w, (x, y))]]$ with $\ell : \mathbb{R}^M \times (X \times Y) \mapsto \mathbb{R}$.

**Lemma 5.** *If (i)* $\ell(w, (x, y))$ *is continuously differentiable in* $w$ *up to order 2, and (ii)* $\lambda_{max}(\nabla_w^2\ell(w, (x, y))) \leq B_H$, *then for* $\hat{w} \in \mathbb{R}^M$ *and* $q(w) = \mathcal{N}(w|\hat{w}, \epsilon I)$

$$r_{Gib}(q(w)) = \underset{(x,y)\sim D}{\text{E}}[\underset{q(w)}{\text{E}}[\ell(w, (x, y))]] \leq r_{Gib}(\delta(w - \hat{w})) + \frac{1}{2}\epsilon MB_H. \quad (7)$$

*Proof.* By the multivariable Taylor's theorem, for $\hat{w} \in \mathbb{R}^M$

$$\ell(w, (x, y)) = \ell(\hat{w}, (x, y)) + \left(\nabla_w\ell(w, (x, y))\Big|_{w=\hat{w}}\right)^T (w - \hat{w})$$

$$+ \frac{1}{2}(w - \hat{w})^T\left(\nabla_w^2\ell(w, (x, y))\Big|_{w=\tilde{w}}\right)(w - \hat{w})$$

where $\nabla_w\ell(w, (x, y))$ and $\nabla_w^2\ell(w, (x, y))$ denote the gradient and Hessian, and $\tilde{w} = (1 - \alpha)\hat{w} + \alpha w$ for some $\alpha \in [0, 1]$ where $\alpha$ is a function of $w$. Taking the expectation results in

$$\underset{q(w)}{\text{E}}[\ell(w, (x, y))] = \ell(\hat{w}, (x, y)) + \frac{1}{2}\underset{q(w)}{\text{E}}[(w - \hat{w})^T\nabla_w^2\ell(w, (x, y))\Big|_{w=\tilde{w}}(w - \hat{w})]. \quad (8)$$

If the maximum eigenvalue of $\nabla_w^2 \ell(w, (x, y))$ is bounded uniformly by some $B_H < \infty$, then the second term of (8) is bounded above by $\frac{1}{2} B_H \mathrm{E}[(w - \hat{w})^T (w - \hat{w})] = \frac{1}{2} \epsilon M B_H$. Taking expectation w.r.t. $D$ yields the statement of the lemma. $\qquad\square$

Since $Q$ includes $\epsilon$-inflated distributions centered on $\hat{w}$ where $\|\hat{w}\|_2 \leq B_m$, we have the following.

**Theorem 6 (Bound on Gibbs Risk Against Point Estimate Competitors).** *If (i)* $-\log \mathrm{E}_{p(f|w)}[p(y|f)]$ *is continuously differentiable in* $w$ *up to order 2, and (ii)* $\lambda_{max}\left(\nabla_w^2\left(-\log \mathrm{E}_{p(f|w)}[p(y|f)]\right)\right) \leq B_H$, *then, for all* $\hat{w}$ *with* $\|\hat{w}\|_2 \leq B_m$,

$$\mathrm{E}_{S \sim D^n}[r_{2Gib}(q_{2Bi}^\star(w))] \leq r_{2Gib}\left(\delta\left(w - \hat{w}\right)\right) + \Delta(B_H) + \frac{1}{\lambda}\Psi(\lambda, n),$$

$$\Delta(B_H) \triangleq \frac{1}{2}M\left(\frac{1}{n} + \frac{1}{\lambda}\right)\left(\frac{2}{M}B_R' + 1 + \log\left(B_H\left(\frac{n\lambda}{n+\lambda}\right)\right)\right). \quad (9)$$

*Proof.* Using the distribution $q = \mathcal{N}(w|\hat{w}, \epsilon I)$ in the RHS of Corollary 3 yields

$$\mathrm{E}_{S \sim D^n}[r_{2Gib}(q_{2Bi}^\star(w))] \leq r_{2Gib}(q) + \frac{1}{\eta n}\mathrm{KL}\left(q\|p\right) + \frac{1}{\lambda}\max_{q \in Q}\mathrm{KL}\left(q\|p\right) + \frac{1}{\lambda}\Psi(\lambda, n)$$

$$\leq r_{2Gib}\left(\delta\left(w - \hat{w}\right)\right) + \frac{1}{2}\epsilon M B_H - \frac{1}{2}AM\log\epsilon + AB_R' + \frac{1}{\lambda}\Psi(\lambda, n) \quad (10)$$

where $A = \left(\frac{1}{\eta n} + \frac{1}{\lambda}\right)$ and we have used Lemma 4 and Lemma 5. Eq (10) is optimized when $\epsilon = \frac{A}{B_H}$. Re-substituting the optimal $\epsilon$ in (10) yields

$$\mathrm{E}_{S \sim D^n}[r_{2Gib}(q_{2Bi}^\star(w))] \leq r_{2Gib}\left(\delta\left(w - \hat{w}\right)\right)$$

$$+ \frac{1}{2}M\left(\frac{1}{\eta n} + \frac{1}{\lambda}\right)\left(\frac{2}{M}B_R' + 1 - \log\left(\frac{1}{B_H}\left(\frac{1}{\eta n} + \frac{1}{\lambda}\right)\right)\right) + \frac{1}{\lambda}\Psi(\lambda, n). \quad (11)$$

Setting $\eta = 1$ yields the result. $\qquad\square$

The theorem calls for running the variational algorithm with constraints on eigenvalues of $V$. The fixed-point characterization [21] of the optimal solution in *linear LGM* implies that such constraints hold for the optimal solution. Therefore, they need not be enforced explicitly in these models.

For any distribution $q(w)$ and function $f(w)$ we have $\min_w [f(w)] \leq \mathrm{E}_{q(w)}[f(w)]$. Therefore, the minimizer of the Gibbs risk is a point estimate, which with Theorem 6 implies:

**Corollary 7.** *Under the conditions of Theorem 6, for all* $q(w) = \mathcal{N}(w|m, V)$ *with* $\|m\|_2 \leq B_m$, $\mathrm{E}_{S \sim D^n}[r_{2Gib}(q_{2Bi}^\star(w))] \leq r_{2Gib}\left(q(w)\right) + \Delta(B_H) + \frac{1}{\lambda}\Psi(\lambda, n)$.

More importantly, as another immediate corollary, we have a bound for the Bayes risk:

**Corollary 8 (Bound on Bayes Risk Against Point Estimate Competitors).** *Under the conditions of Theorem 6, for all* $\hat{w}$ *with* $\|\hat{w}\|_2 \leq B_m$,

$\mathrm{E}_{S \sim D^n}[r_{2Bay}(q_{2Bi}^\star(w))] \leq r_{2Bay}\left(\delta\left(w - \hat{w}\right)\right) + \Delta(B_H) + \frac{1}{\lambda}\Psi(\lambda, n)$.

*Proof.* Follows from (a) $\forall q, r_{2Bay}(q) \leq r_{2Gib}(q)$ (Jensen's inequality), and (b) $\forall \hat{w} \in \mathbb{R}^M, r_{2Bay}(\delta(w - \hat{w})) = r_{2Gib}(\delta(w - \hat{w}))$. $\qquad\square$

The extension for Bayes risk in step b of the proof is only possible thanks to the extension to point estimates. As stated in the previous section, for bounded losses, $\Psi(\lambda, n)$ is bounded as $\frac{\lambda^2(b-a)^2}{2n}$. As in [7], we can choose $\lambda = \sqrt{n}$ or $\lambda = n$ to obtain decays rates $\frac{\log n}{\sqrt{n}}$ or $\frac{\log n}{n}$ respectively, where the latter has a fixed non-decaying gap term $(b-a)^2/2$. However, unlike [7], in our proof both cases are achievable with $\eta = 1$, i.e., for the variational algorithm. For example,

using $\eta = 1$, $\lambda = \sqrt{n}$, the prior with $\mu = 0$ and $\Sigma = \frac{1}{M}(MB_V + B_m^2)I$, and bounded loss,

$$\Delta(B_H) + \frac{1}{\lambda}\Psi(\lambda, n) \le \frac{M}{\sqrt{n}}\left(1 + \log B_H + \log n + \log\left(B_V + \frac{1}{M}B_m^2\right) + \frac{(b-a)^2}{2M}\right).$$

The results above are developed for the log loss but we can apply them more generally. Toward this we note that Corollary 3 holds for an arbitrary loss, and Lemma 5, and Theorem 6 hold for a sufficiently smooth loss with bounded 2nd derivative w.r.t. $w$. The conversion to Bayes risk in Corollary 8 holds for any loss convex in $p$. Therefore, the result of Corollary 8 holds more generally for any sufficiently smooth loss that has bounded 2nd derivative in $w$ and that is convex in $p$. We provide an application of this more general result in the next section.

## 4.2 Applications in Concrete Models

This section develops bounds on $\Psi$ and $B_H$ for members of the 2L family.

**CTM:** For a document, the generative model for CTM first draws $w \sim \mathcal{N}(\mu, \Sigma)$, $w \in \mathbb{R}^{K-1}$ where $\{\mu, \Sigma\}$ are model parameters, and then maps this vector to the $K$-simplex with the logistic transformation, $\theta = h(w)$. For each position $i$ in the document, the latent topic variable, $f_i$, is drawn from Discrete$(\theta)$, and the word $y_i$ is drawn from a Discrete$(\beta_{f_i, \cdot})$ where $\beta$ denotes the topics and is treated as a parameter of the model. In this case $p(f|w)$ can be integrated out analytically and the loss is $-\log\left(\sum_{k=1}^{K}\beta_{k,y}h_k(w)\right)$. We have (proof in supplementary material):

**Corollary 9.** *For CTM models where the parameters $\beta_{k,y}$ are uniformly bounded away from $0$, i.e., $\beta_{k,y} \ge \gamma > 0$, for all $\hat{w}$ with $\|\hat{w}\|_2 \le B_m$,*

$$\mathrm{E}_{S \sim D^n}[r_{2Bay}(q_{2Bi}^\star(w))] \le r_{2Bay}\left(\delta(w - \hat{w})\right) + \Delta(B_H) + \frac{\lambda(\log\gamma)^2}{2n} \text{ with } B_H = 5.$$

The following lemma is expressed in terms of log loss but also holds for smoothed log loss (proof in supplementary material):

**Lemma 10.** *When $f$ is a deterministic function of $w$, if (i) $-\log p\left(y|f(w,x)\right)$ is continuously differentiable in $f$ up to order 2, and $f(w,x)$ is continuously differentiable in $w$ up to order 2, (ii) $\frac{\partial^2\left[-\log p(y|f)\right]}{\partial f^2} \le c_2$, (iii) $\left|\frac{\partial\left[-\log p(y|f)\right]}{\partial f}\right| \le c_1$, (iv) $\|\nabla_w f(w,x)\|_2^2 \le c_1^f$, and (v) $\sigma_{max}\left(\nabla_w^2 f(w,x)\right) \le c_2^f$ ($\sigma_{max}$ is the max singular value), then $B_H = c_2 c_1^f + c_1 c_2^f$.*

**GLM:** The bound of [11] for GLM was developed for exact Bayesian inference. The following corollary extends this to approximate inference through RCLM. In GLM, $f = w^T x$, $\|\nabla_w\|^2 = \|x\|^2$, and $\nabla_w^2 = 0$ and a bound on $B_H$ is immediate from Lemma 10. In addition the smoothed loss is bounded $0 \le \tilde{\ell} \le -\log\alpha$. This implies

**Corollary 11.** *For GLM, if (i) $\tilde{\ell}(w,(x,y)) = -\log((1-\alpha)p\left(y|f(w,x)\right) + \alpha)$ is continuously differentiable in $f$ up to order 2, and (ii) $\frac{\partial^2\tilde{\ell}}{\partial f^2} \le c$, then, for all $\hat{w}$ with $\|\hat{w}\|_2 \le B_m$,*

$$\mathrm{E}_{S \sim D^n}[\tilde{r}_{2Bay}(\tilde{q}_{2Bi}^\star(w))] \le \tilde{r}_{2Bay}\left(\delta(w - \hat{w})\right) + \Delta(B_H) + \frac{\lambda(\log\alpha)^2}{2n} \text{ with } B_H = c\max_{x \in X}\|x\|_2^2.$$

We develop the bound $c$ for the logistic and Normal likelihoods (see supplementary material). Let $\alpha' = \frac{\alpha}{1-\alpha}$. For the logistic likelihood $\sigma(yf)$, we have $c = \frac{1}{16}\frac{1}{(\alpha')^2} + \frac{\sqrt{3}}{18}\frac{1}{\alpha'}$. For the Gaussian likelihood $\frac{1}{\sqrt{2\pi}\sigma_Y}\exp(-\frac{1}{2}\frac{(y-f)^2}{\sigma_Y^2})$, we have $c = \frac{1}{2\pi\sigma_Y^4 e}\frac{1}{(\alpha')^2} + \frac{1}{\sqrt{2\pi}\sigma_Y^3}\frac{1}{\alpha'}$.

The work of [7] has claimed[3] a bound on the Gibbs risk for linear regression which should be compared to our result for the Gaussian likelihood. Their result is developed under the assumption that the Bayesian model specification is correct and in addition that $x$ is generated from $x \sim \mathcal{N}(0, \sigma_x^2 I)$. In contrast our result, using the smoothed loss, holds for arbitrary distributions $D$ without the assumption of correct model specification.

**Sparse GP:** In the sparse GP model, the conditional is $p\left(f|w,x\right) = \mathcal{N}(f|a(x)^T w + b(x), \sigma^2(x))$ where $a(x)^T = K_{Ux}^T K_{UU}^{-1}$, $b(x) = \mu_x - K_{Ux}^T K_{UU}^{-1}\mu_U$ and $\sigma^2(x) = K_{xx} - K_{Ux}^T K_{UU}^{-1} K_{Ux}$ with $\mu$ denoting the mean function and $K_{Ux}$, $K_{UU}$ denoting the kernel matrix evaluated at inputs $(U,x)$ and $(U,U)$ respectively. In the conjugate case, the likelihood is given by $p\left(y|f\right) = \mathcal{N}(y|f,\sigma_Y^2)$ and integrating $f$ out yields $\mathcal{N}(y|a(x)^T w + b(x), \sigma^2(x) + \sigma_Y^2)$. Using the smoothed loss, we obtain:

**Corollary 12.** *For conjugate sparse GP, for all $\hat{w}$ with $\|\hat{w}\|_2 \le B_m$,*

$\mathrm{E}_{S \sim D^n}[\tilde{r}_{2Bay}(\tilde{q}_{2Bi}^\star(w))] \le \tilde{r}_{2Bay}\left(\delta\left(w - \hat{w}\right)\right) + \Delta(B_H) + \frac{\lambda(\log\alpha)^2}{2n}$ *with* $B_H = c\max_{x \in X}\|a(x)\|_2^2$, *where* $c = \frac{1}{2\pi\sigma_Y^4 e}\frac{1}{(\alpha')^2} + \frac{1}{\sqrt{2\pi}\sigma_Y^3}\frac{1}{\alpha'}$.

*Proof.* The Hessian is given by $\nabla_w^2\tilde{\ell}(w,(x,y)) = \frac{1}{(\mathcal{N}+\alpha')^2}\nabla_w\mathcal{N}(\nabla_w\mathcal{N})^T - \frac{1}{\mathcal{N}+\alpha'}\nabla_w^2\mathcal{N}$ where $\mathcal{N}$ denotes $\mathcal{N}(y|f(w),\sigma^2(x) + \sigma_Y^2)$, with $f(w) = a(x)^T w + b(x)$. The gradient $\nabla_w\mathcal{N}$ equals $\left(\frac{\partial\mathcal{N}}{\partial(f(w))}\right)a(x)$ and the Hessian $\nabla_w^2\mathcal{N}$ equals $\left(\frac{\partial^2\mathcal{N}}{\partial(f(w))^2}\right)a(x)a(x)^T$. Therefore, $\nabla_w^2\tilde{\ell} = \left(\frac{1}{(\mathcal{N}+\alpha')^2}\left(\frac{\partial\mathcal{N}}{\partial(f(w))}\right)^2 - \frac{1}{\mathcal{N}+\alpha'}\frac{\partial^2\mathcal{N}}{\partial(f(w))^2}\right)a(x)a(x)^T = \frac{\partial^2\left[-\log\left((1-\alpha)\mathcal{N}+\alpha\right)\right]}{\partial(f(w))^2}a(x)a(x)^T$. The result of Corollary 11 for Gaussian likelihood can be used to bound the 2nd derivative of the smoothed loss: $\frac{\partial^2\left[-\log\left((1-\alpha)\mathcal{N}+\alpha\right)\right]}{\partial(f(w))^2} \le \frac{1}{2\pi(\sigma^2(x)+\sigma_Y^2)^2 e}\frac{1}{(\alpha')^2} + \frac{1}{\sqrt{2\pi}(\sigma^2(x)+\sigma_Y^2)^{\frac{3}{2}}}\frac{1}{\alpha'} \le \frac{1}{2\pi\sigma_Y^4 e}\frac{1}{(\alpha')^2} + \frac{1}{\sqrt{2\pi}\sigma_Y^3}\frac{1}{\alpha'} = c$. Finally, the eigenvalue of the rank-1 matrix $ca(x)a(x)^T$ is bounded by $c\max_{x \in X}\|a(x)\|_2^2$. $\qquad\square$

**Remark 1.** We noted above that, for sGP, $q_{2Bi}^\star$ does not correspond to a variational algorithm. The standard variational approach uses $q_{2A}^\star$ and the collapsed bound uses $q_{2Bj}^\star$ (but requires cubic time). It can be shown that $q_{2Bi}^\star$ corresponds exactly to the fully independent training conditional (FITC) approximation for sGP [24, 16] in that their optimal solutions are identical. Our result can be seen to justify the use of this algorithm which is known to perform well empirically.

Finally, we consider binary classification in GLM with the convex loss function $\ell'(w,(x,y)) = \frac{1}{8}(y - (2p(y|w,x) - 1))^2$. The proof of the following corollary is in the supplementary material:

**Corollary 13.** *For GLM with $p(y|w,x) = \sigma(yw^T x)$, for all $\hat{w}$ with $\|\hat{w}\|_2 \le B_m$,* $\mathrm{E}_{S \sim D^n}[r_{2Bay}'(q_{2Bi}'^\star(w))] \le r_{2Bay}'\left(\delta\left(w - \hat{w}\right)\right) + \Delta(B_H) + \frac{\lambda}{8n}$ *with $B_H = \frac{5}{16}\max_{x \in X}\|x\|_2^2$.*

### 4.3 Direct Application of RCLM to Conjugate Linear LGM

In this section we derive a bound for an algorithm that optimizes a surrogate of the loss directly. In particular, we consider the Bayes loss for linear LGM with conjugate likelihood $p(y|f) = \mathcal{N}(y|f,\sigma_Y^2)$ where $-\log\mathrm{E}_{q(w)}[\mathrm{E}_{p(f|w)}[p(y|f)]] = -\log\mathcal{N}(y|a^T m + b, \sigma^2 + \sigma_Y^2 + a^T V a)$ and where $a, b,$ and $\sigma^2$ are functions of $x$. This includes, for example, linear regression and conjugate sGP.

The proposed algorithm $q_{2Ds}^\star$ performs RCLM with competitor set $\Theta = \{(m,V) : \|m\|_2 \le B_m, V \in \mathbb{S}_{++}, \|V\|_F \le B_V\}$, regularizer $R(m,V) = \frac{1}{2}\|m\|_2^2 + \frac{1}{2}\|V\|_F^2$, $\eta = \frac{1}{\sqrt{n}}$ and the surrogate loss $\ell^{surr}(m,V) = \frac{1}{2}\log(2\pi) + \frac{1}{2}\left(\sigma^2 + \sigma_Y^2 + a^T V a\right) + \frac{1}{2}\frac{(y - a^T m - b)^2}{\sigma^2 + \sigma_Y^2 + a^T V a}$. With these definitions we can apply Theorem 1 to get (proof in supplementary material):

**Theorem 14.** *With probability at least $1 - \delta$, $r_{2Bay}(q_{2Ds}^\star) \le \min_{q \in Q} r_{2Bay}^{surr}(q(w)) + \frac{1}{\delta\sqrt{n}}\left(B_m^2 + B_V^2 + 8(\rho_m^2 + \rho_V^2)\right)$ where $\rho_m = \frac{1}{\sigma_Y^2}\max_{x \in X}\|a\|_2\max_{x \in X, y \in Y, m}|y - a^T m - b|$ and $\rho_V = \frac{1}{2\sigma_Y^2}\max_{x \in X, y \in Y, m}\|a\|_2^2\left(1 + \frac{(y - a^T m - b)^2}{\sigma_Y^2}\right)$.*

## 5 Direct Loss Minimization

The results in this paper expose the fact that different algorithms are apparently implicitly optimizing criteria for different loss functions. In particular, $q_{2A}^\star$ optimizes for $r_{2A}$, $q_{2Bi}^\star$ optimizes for $r_{2Gib}$

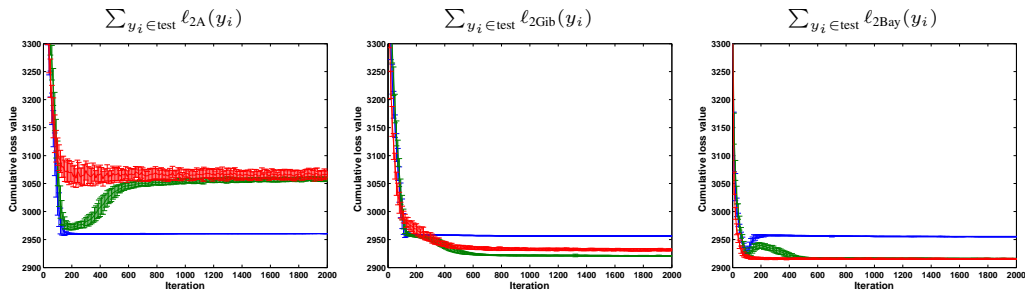

Figure 1: *Artificial data.* Cumulative test set losses of different variational algorithms. x-axis is iteration. Mean $\pm\, 1\sigma$ of 30 trials are shown per objective. $q_{2A}^\star$ in blue. $q_{2Bi}^\star$ in green. $q_{2D}^\star$ in red.

and $q_{2D}^\star$ optimizes for $r_{2Bay}$. Even though we were able to bound $r_{2Bay}$ of the $q_{2Bi}^\star$ algorithm, it is interesting to check the performance of these algorithms in practice.

We present an experimental study comparing these algorithms on the correlated topic model (CTM) that was described in the previous section. To explore the relation between the algorithms and their performance we run the three algorithms and report their empirical risk on a test set, where the risk is also measured in three different ways. Figure 1 shows the corresponding learning curves on an artificial document generated from the model. Full experimental details and additional results on a real dataset are given in the supplementary material.

We observe that at convergence each algorithm is best at optimizing its own implicit criterion. However, considering $r_{2Bay}$, the differences between the outputs of the variational algorithm $q_{2Bi}^\star$ and direct loss minimization $q_{2D}^\star$ are relatively small. We also see that at least in this case $q_{2Bi}^\star$ takes longer to reach the optimal point for $r_{2Bay}$. Clearly, except for its own implicit criterion, $q_{2A}^\star$ should not be used. This agrees with prior empirical work on $q_{2A}^\star$ and $q_{2Bi}^\star$ [22]. The current experiment shows the potential of direct loss optimization for improved performance but justifies the use of $q_{2Bi}^\star$ both under correct model specification (artificial data) and when the model is incorrect (real data in supplement).

Preliminary experiments in sparse GP show similar trends. The comparison in that case is more complex because $q_{2Bi}^\star$ is not the same as the collapsed variational approximation, which in turn requires cubic time to compute, and we additionally have the surrogate optimizer $q_{2Ds}^\star$. We defer a full empirical exploration in sparse GP to future work.

## 6   Discussion

The paper provides agnostic learning bounds for the risk of the Bayesian predictor, which uses the posterior calculated by RCLM, against the best single predictor. The bounds apply for a wide class of Bayesian models, including GLM, sGP and CTM. For CTM our bound applies precisely to the variational algorithm with the collapsed variational bound. For sGP and GLM the bounds apply to bounded variants of the log loss. The results add theoretical understanding of why approximate inference algorithms are successful, even though they optimize the wrong objective, and therefore justify the use of such algorithms. In addition, we expose a discrepancy between the loss used in optimization and the loss typically used in evaluation and propose alternative algorithms using regularized loss minimization. A preliminary empirical evaluation in CTM shows the potential of direct loss minimization but that the collapsed variational approximation $q_{2Bi}^\star$ has the advantage of strong theoretical guarantees and excellent empirical performance, both when the Bayesian model is correct and under model misspecification.

Our results can be seen as a first step toward full analysis of approximate Bayesian inference methods. One limitation is that the competitor class in our results is restricted to point estimates. While point estimate predictors are optimal for the Gibbs risk, they are not optimal for Bayes predictors. In addition, the bounds show that the Bayesian procedures will do almost as well as the best point estimator. However, they do not show an advantage over such estimators, whereas one would expect such an advantage. It would also be interesting to incorporate direct loss minimization within the Bayesian framework. These issues remain an important challenge for future work.

## Acknowledgments

This work was partly supported by NSF under grant IIS-1714440.

## Footnotes

[1]For the smoothed log loss, the translation can be applied prior to the re-scaling, i.e., $-\log(\frac{1-\alpha}{\max_{w,x,y} p(y|w,x)}p(y|w, x) + \alpha)$.

[2] [20] analyzed regularized average loss but the same proof steps with minor modifications yield the statement for cumulative loss given here.

[3] Denoting $\Delta r_i(w) = r_W(w) - \hat{r}_W(w,(x_i,y_i))$ and $f_i(w,n,\lambda) = \mathrm{E}_{p(\Delta r_i(w))}[\exp\left(\frac{\lambda}{n}\Delta r_i(w)\right)]$, the proof of Corollary 5 in [7] erroneously replaces $\mathrm{E}_{p(w)}[\prod_i f_i(w,n,\lambda)]$ with $\prod_i \mathrm{E}_{p(w)}[f_i(w,n,\lambda)]$. We are not aware of a correction of this proof which yields a correct bound for $\Psi$ without using a smoothed loss. Any such bound would, of course, be applicable with our Corollary 8.

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
