[Supplementary Material]

# Supplementary material for "Excess risk bounds for the Bayes risk using variational inference in latent Gaussian models"

## A    Details for Section 3

**Convexity Based results from RCLM Theorem:** We have already noted that $q_{\text{Gib}}^\star(w)$ is the RCLM solution for $\ell_{\text{Gib}}(q(w),(x,y)) = -\operatorname{E}_{q(w)}[\log p(y|w,x)]$. We therefore get:

**Corollary 15.** *If the $KL$ regularizer is $\sigma$-strong-convex over $Q$ and $\ell_{Gib}(q(w),(x,y))$ is $\rho$-Lipschitz and convex in $q(w)$, then, for all $q \in Q$, $\operatorname{E}_{S \sim D^n}[r_{Gib}(q_{Gib}^\star(w))] \leq r_{Gib}(q) + \frac{1}{\eta n} KL\left(q\|p\right) + \frac{4\rho^2 \eta}{\sigma}$.*

Similarly, $q_{\text{2Bi}}^\star(w)$ and $q_{\text{2D}}^\star(w)$ are RCLM solutions for the 2L model with losses defined as $\ell_{\text{2Gib}}(q(w),(x,y)) = -\operatorname{E}_{q(w)}[\log \operatorname{E}_{p(f|w,x)}[p(y|f)]]$ and $\ell_{\text{2Bay}}(q(w),(x,y)) = -\log \operatorname{E}_{q(w)}[\operatorname{E}_{p(f|w,x)}[p(y|f)]]$. We therefore get:

**Corollary 16.** *If the $KL$ regularizer is $\sigma$-strong-convex over $Q$ and $\ell_{2Gib}(q(w),(x,y))$ is $\rho$-Lipschitz and convex in $q(w)$ then, for all $q \in Q$, $\operatorname{E}_{S \sim D^n}[r_{2Gib}(q_{2Bi}^\star(w))] \leq r_{2Gib}(q) + \frac{1}{\eta n} KL(q\|p) + \frac{4\rho^2 \eta}{\sigma}$.*

**Corollary 17.** *If the $KL$ regularizer is $\sigma$-strong-convex over $Q$ and $\ell_{2Bay}(q(w),(x,y))$ is $\rho$-Lipschitz and convex in $q(w)$ then, for all $q \in Q$, $\operatorname{E}_{S \sim D^n}[r_{2Bay}(q_{2D}^\star(w))] \leq r_{2Bay}(q) + \frac{1}{\eta n} KL(q\|p) + \frac{4\rho^2 \eta}{\sigma}$.*

Note that for the variational algorithm $\eta = 1$ and the KL term decays at a rate of $1/n$ but the residual term $\frac{4\rho^2 \eta}{\sigma}$ does not decay. On the other hand if we use $\eta = 1/\sqrt{n}$ both terms decay at a rate of $1/\sqrt{n}$. This type of behavior was pointed out by [7] and holds for the result of [1]. The main paper discusses how Theorem 2 achieves both rates using $\eta = 1$.

**Proof of Theorem 2 (RCLM Theorem):** We start with a lemma from [20]. Recall that for hypothesis space $H$ and hypothesis $h \in H$ we used the notation $\ell(h,(x,y))$, and $r(h) = \operatorname{E}_{(x,y)\sim D}[\ell(h,(x,y))]$ and defined the solution $\text{RCLM}(H,\ell,R,\eta,S)$. Similarly for a sample $S$ define $\hat{r}(h,S) = \frac{1}{n}\sum_i \ell\left(h,(x_i,y_i)\right)$.

**Lemma 18** ([20]). *If $h^\star = RCLM(H,\ell,R,\eta,S)$, then $\forall h \in H$,*

$$\operatorname*{E}_{S \sim D^n}[r(h^\star)] \leq r(h) + \frac{1}{\eta n} R(h) + \operatorname*{E}_{S \sim D^n}[r(h^\star) - \hat{r}(h^\star,S)]. \tag{12}$$

For completeness we include the proof for this lemma [20]:

*Proof.* We prove the equivalent statement $\forall q \in Q, \operatorname{E}_{S \sim D^n}[\hat{r}(q^\star,S)] \leq r(q) + \frac{1}{\eta n} R(q)$. Since $q^\star = \text{RCLM}(Q,\ell,R,\eta,S)$, $\forall q \in Q, n\hat{r}(q,S) + \frac{1}{\eta} R(q) \geq n\hat{r}(q^\star,S) + \frac{1}{\eta} R(q^\star) \geq n\hat{r}(q^\star,S)$. Dividing both sides by $n$ and taking expectations with respect to $D^n$ yields the result. $\square$

We next extend the reasoning from [7] to derive a bound for RCLM. For this recall the notation of loss and risk for individual base parameters $\ell_W(w,(x,y)) = -\log p(y|w,x)$, $r_W(w) = \operatorname{E}_D[\ell_W(w,(x,y))]$, and $\hat{r}_W(w,S) = -\frac{1}{n}\sum_i \log p(y_i|w,x_i)$.

The compression lemma [2] states that for any measurable function $f$, we have $E_{q(w)}[f(w)] \leq KL(q(w)\|p(w)) + \log E_{p(w)}[e^{f(w)}]$. As in [7], applying this to the function, $\lambda(r_W(w) - \hat{r}_W(w,S))$ the following is valid for all $q$:

$$r_{\text{Gib}}(q(w)) - \hat{r}_{\text{Gib}}(q(w),S) \leq \frac{1}{\lambda}\left(\text{KL}\left(q\|p\right) + \log \operatorname*{E}_{p(w)}[e^{\lambda\left(r_W(w) - \hat{r}_W(w,S)\right)}]\right) \tag{13}$$

where $\lambda$ is a scalar. Evaluating (13) at $q^\star \in Q$ (note that $q^\star$ is a function of the sample $S$) and taking expectations of both sides with respect to the draw of the sample yields

$$\underset{S \sim D^n}{\mathrm{E}}[r_{\mathrm{Gib}}(q^\star) - \hat{r}_{\mathrm{Gib}}(q^\star, S)] \leq \underset{S \sim D^n}{\mathrm{E}}[\frac{1}{\lambda}\left(\mathrm{KL}\left(q^\star\|p\right) + \log \underset{p(w)}{\mathrm{E}}[e^{\lambda\left(r_W(w) - \hat{r}_W(w,S)\right)}]\right)]$$

$$= \frac{1}{\lambda}\left(\underset{S \sim D^n}{\mathrm{E}}[\mathrm{KL}\left(q^\star\|p\right)] + \underset{S \sim D^n}{\mathrm{E}}[\log \underset{p(w)}{\mathrm{E}}[e^{\lambda\left(r_W(w) - \hat{r}_W(w,S)\right)}]]\right)$$

$$\leq \frac{1}{\lambda}\left(\max_{q \in Q}\mathrm{KL}\left(q\|p\right) + \underset{S \sim D^n}{\mathrm{E}}[\log \underset{p(w)}{\mathrm{E}}[e^{\lambda\left(r_W(w) - \hat{r}_W(w,S)\right)}]]\right)$$

$$\leq \frac{1}{\lambda}\left(\max_{q \in Q}\mathrm{KL}\left(q\|p\right) + \log \underset{S \sim D^n}{\mathrm{E}}[\underset{p(w)}{\mathrm{E}}[e^{\lambda\left(r_W(w) - \hat{r}_W(w,S)\right)}]]\right)$$

$$= \frac{1}{\lambda}\left(\max_{q \in Q}\mathrm{KL}\left(q\|p\right) + \Psi(\lambda, n)\right) \tag{14}$$

where in the third step we replace expectation with maximum, in the fourth step we used Jensen's inequality, and $\Psi(\lambda, n) = \log \mathrm{E}_{S \sim D^n}[\mathrm{E}_{p(w)}[e^{\lambda\left(r_W(w) - \hat{r}_W(w,S)\right)}]]$ as defined in [7].

Applying Lemma 18 to the RCLM algorithm $q^\star_{\mathrm{Gib}}$ and combining this with (14) we get that for all $q \in Q$:

$$\underset{S \sim D^n}{\mathrm{E}}[r_{\mathrm{Gib}}(q^\star)] \leq r_{\mathrm{Gib}}(q) + \frac{1}{\eta n}\mathrm{KL}\left(q\|p\right) + \frac{1}{\lambda}\max_{q \in Q}\mathrm{KL}\left(q\|p\right) + \frac{1}{\lambda}\Psi(\lambda, n). \tag{15}$$

$\square$

## A.1 Bounds on $\Psi(\lambda, n)$

For completeness this subsection recalls facts from [4, 7] who provided bounds on $\Psi(\lambda, n)$.

Let $\hat{r}_W(w, (x_i, y_i))$ denote $-\log p(y_i|w, x_i)$. By considering the loss centered w.r.t. $D$, $\Delta r_i(w) = r_W(w) - \hat{r}_W(w, (x_i, y_i))$, $\Psi(\lambda, n)$ can be expressed as[4]

$$\Psi(\lambda, n) = \log \underset{p(w)}{\mathrm{E}}[\prod_i \underset{p(\Delta r_i(w))}{\mathrm{E}}[\exp\left(\frac{\lambda}{n}\Delta r_i(w)\right)]]. \tag{16}$$

Each term of the product in (16) is the moment generating function (MGF) of $\Delta r_i(w)$. Hence, if the MGF of $\Delta r_i(w)$ can be bounded, then (16) can be evaluated.

If $-\log p(y|w, x)$ is bounded in $[a, b]$, then $\Delta r_i(w)$ will be bounded in $[a - b, b - a]$ (and zero-mean). Then, by Hoeffding's lemma [4], $\mathrm{E}_{p(\Delta r_i(w))}[\exp\left(\frac{\lambda}{n}\Delta r_i(w)\right)] \leq \exp\left(\frac{\lambda^2(b-a)^2}{2n^2}\right)$. Plugging this into (16) we observe that the expectation over $p(w)$ returns the same bound (because its argument does not depend on $w$) and we get $\Psi(\lambda, n) \leq \frac{\lambda^2(b-a)^2}{2n}$. If $-\log p(y|w, x)$ is not bounded, but $\Delta r_i(w)$ is sub-Gaussian or sub-gamma, then $\Psi(\lambda, n)$ can be bounded with additional assumptions on the underlying distribution $D$.

## B Details for Section 4

The KL divergence from $q(w) = \mathcal{N}(w|m, V)$ to $p(w) = \mathcal{N}(w|\mu, \Sigma)$ is given by

$$\mathrm{KL}\left(q\|p\right) = \frac{1}{2}\left(\mathrm{tr}(\Sigma^{-1}V) + (\mu - m)^T\Sigma^{-1}(\mu - m) + \log\frac{|\Sigma|}{|V|} - M\right). \tag{17}$$

We first show that the KL divergence is $\sigma$ strong convex (needed for Corollaries 15, 16, 17).

**Lemma 19.** *If (i)* $b_V \leq \lambda_j(V) \leq B_V$ *for* $1 \leq j \leq M$*, and (ii)* $\|m\|_2 \leq B_m$*, then* $KL\left(\mathcal{N}(m,V)\|\mathcal{N}(\mu,\Sigma)\right)$ *is* $\sigma$*-strong convex w.r.t. vector and matrix* $\|\cdot\|_2$*, where* $\sigma = \min\left(\frac{1}{\lambda_{max}(\Sigma)}, \frac{1}{2B_V^2}\right)$,

*Proof.* Since $\nabla_m^2 KL(m,V) = \Sigma^{-1}$, (17) is $\sigma_m$-strongly convex w.r.t. $m$ where $\sigma_m = \frac{1}{\lambda_{max}(\Sigma)}$. W.r.t. $V$, $\nabla_V^2 KL(m,V) = \frac{1}{2}V^{-1} \otimes V^{-1}$, and (17) is not strongly convex unless its minimum eigenvalue $\frac{1}{2(\lambda_{max}(V))^2}$ is bounded away from zero. The condition $\frac{1}{2(\lambda_{max}(V))^2} \geq \sigma_V > 0$ requires $\lambda_{max}(V) \leq \frac{1}{\sqrt{2\sigma_V}} \triangleq B_V$. Hence, $\sigma = \min(\sigma_m, \sigma_V) = \min\left(\frac{1}{\lambda_{max}(\Sigma)}, \frac{1}{2B_V^2}\right)$. $\qquad\square$

**Proof of Lemma 4 (KL Bound):** Let $\lambda_{min}(V) \geq b_V$ and $\lambda_{max}(V) = \|V\|_2 \leq B_V$. Use $\text{tr}(AB) \leq \lambda_{max}(B)\,\text{tr}(A)$ (see e.g., [27]) to bound the first term of (17) with $\frac{MB_V}{\lambda_{min}(\Sigma)}$. Bound the second term of (17) w.r.t. 2-norm as $\|\mu - m\|_2^2 \|\Sigma^{-1}\|_2 \leq \left(\|\mu\|_2^2 + \|m\|_2^2\right)\lambda_{max}\left(\Sigma^{-1}\right) = \frac{\|\mu\|_2^2 + \|m\|_2^2}{\lambda_{min}(\Sigma)}$. Finally, for a symmetric positive definite matrix $A$ of dimension $M$, use the identity $\lambda_{min}(A)^M \leq \det(A) \leq \lambda_{max}(A)^M$ to bound the third term by $M \log \frac{\lambda_{max}(\Sigma)}{b_V}$. $\qquad\square$

**Derivation of** $\frac{1}{\sqrt{n}}$ **rate for Theorem 6 bound:** For $\mu = 0$ and $\Sigma = \nu I$ where $\nu > 0$, $B_R'(\nu) = \frac{1}{2}\left(\frac{1}{\nu}\left(MB_V + B_m^2\right) + M\log\nu - M\right)$. Taking the derivative of $B_R'(\nu)$ w.r.t. $\nu$ yields $\frac{1}{2}\left(-\frac{1}{\nu^2}\left(MB_V + B_m^2\right) + \frac{M}{\nu}\right)$ which equals 0 at $\nu^\star = \frac{1}{M}\left(MB_V + B_m^2\right)$. For this value of $\nu$, we have $B_R'(\nu^\star) = \frac{1}{2}M\log\left(B_V + \frac{1}{M}B_m^2\right)$.

Plugging $B_R'(\nu^\star)$ into the $\Delta(B_H)$ term of (10) yields $\frac{1}{2}AM\left(1 + \log\frac{B_H}{A} + \log\left(B_V + \frac{1}{M}B_m^2\right)\right)$. With $\eta = 1$ and $\lambda = \sqrt{n}$, $A$ equals $\frac{1}{\sqrt{n}} + \frac{1}{n}$, and

$$\Delta(B_H) = \frac{1}{2}\left(\frac{1}{\sqrt{n}} + \frac{1}{n}\right)M\left(1 + \log B_H - \log\left(\frac{1}{\sqrt{n}} + \frac{1}{n}\right) + \log\left(B_V + \frac{1}{M}B_m^2\right)\right)$$

$$\leq \frac{M}{\sqrt{n}}\left(1 + \log B_H + \log n + \log\left(B_V + \frac{1}{M}B_m^2\right)\right).$$

where the inequality follows from the relationship $\frac{2}{n} \leq \frac{1}{\sqrt{n}} + \frac{1}{n} \leq \frac{2}{\sqrt{n}}$. Using the bounded loss result, $\frac{1}{\lambda}\Psi \leq \frac{\lambda(b-a)^2}{2n} = \frac{(b-a)^2}{2\sqrt{n}}$, yields the overall bound:

$$\Delta(B_H) + \frac{1}{\lambda}\Psi(\lambda, n) \leq \frac{M}{\sqrt{n}}\left(1 + \log B_H + \log n + \log\left(B_V + \frac{1}{M}B_m^2\right) + \frac{(b-a)^2}{2M}\right).$$

$\qquad\square$

**Proof of Corollary 9 (CTM):**

We derive bounds on $\Psi()$ and $B_H$ as required in Corollary 8. Recall that CTM uses the logistic transformation $h_k(w) = \frac{\exp(w_k)}{1 + \sum_{l=1}^{K-1}\exp(w_l)}$, if $k < K$ and $h_k(w) = \frac{1}{1 + \sum_{l=1}^{K-1}\exp(w_l)}$, otherwise. The log loss is given by $\ell(w,(x,y)) = -\log\left(\sum_{k=1}^K \beta_{k,y}h_k(w)\right)$. To derive a bound on $\Psi$, note that the loss is bounded since the entries $\beta_{k,y}$ of the topics satisfy $0 < \gamma \leq \beta_{k,y} \leq 1$. Now because $h_k(w)$ is a distribution we have $\gamma \leq \sum_k \beta_{k,y}h_k(w) \leq 1$, and $\Psi \leq \frac{\lambda^2(\log\gamma)^2}{2n}$.

We next show that $B_H$ is bounded. Let $e^{(k)} \in \mathbb{R}^K$ denote the standard Euclidean unit vector in the $k$-th coordinate, and for any $x \in \mathbb{R}^K$, let $\tilde{x} \in \mathbb{R}^{K-1}$ denote the first $K-1$ elements of $x$. Then, the

Hessian of $\ell$ w.r.t. $w$ is described by (suppressing dependence of $\ell$ on $y$ for clarity):

$$
\begin{aligned}
\nabla_w h_k(w) &= h_k(w)(\tilde{e}^{(k)} - \tilde{h}(w)) \\
\nabla_w^2 h_k(w) &= h_k(w)\left[ (\tilde{e}^{(k)} - \tilde{h}(w))(\tilde{e}^{(k)} - \tilde{h}(w))^T - \operatorname{diag}(\tilde{h}(w)) + \tilde{h}(w)\tilde{h}(w)^T \right] \\
\nabla_w \ell(w) &= -\exp(\ell(w))) \sum_k \beta_{k,y} \nabla_w h_k(w) \\
\nabla_w^2 \ell(w) &= -\exp(\ell(w)) \left( \sum_k \beta_{k,y} \nabla_w^2 h_k(w) \right) + \left( \nabla_w \ell(w) \right) \left( \nabla_w \ell(w) \right)^T
\end{aligned}
$$

The maximum eigenvalue of the first term of $\nabla_w^2 \ell(w)$ is

$$
\begin{aligned}
\lambda_{\max}\left( -\exp(\ell(w)) \left( \sum_k \beta_{k,y} \nabla_w^2 h_k(w) \right) \right) &\leq \exp(\ell(w)) \sum_k \beta_{k,y} \lambda_{\max}\left( -\nabla_w^2 h_k(w) \right). \\
&\leq \exp(\ell(w)) \sum_k \beta_{k,y} h_k(w) \max \operatorname{diag}(\tilde{h}(w)) \\
&\leq \exp(\ell(w)) \sum_k \beta_{k,y} h_k(w) \\
&= 1
\end{aligned}
$$

where the first inequality is due to the identity $\lambda_{\max}(A+B) \leq \lambda_{\max}(A) + \lambda_{\max}(B)$ for symmetric $A, B$, the second inequality is due to the previous identity combined with retaining the positive terms of $\lambda_{\max}\left( -\nabla_w^2 h_k(w) \right)$, the third inequality is due to the fact that the elements of $h(w)$ must sum to 1, and the final equality is due to the definition of $\ell(w)$.

The maximum eigenvalue of the second term of $\nabla_w^2 \ell(w)$ is

$$
\begin{aligned}
\lambda_{\max}\left( \left( \nabla_w \ell(w) \right) \left( \nabla_w \ell(w) \right)^T \right) &= \left\| \exp(\ell(w)) \sum_k \beta_{k,y} \nabla_w h_k(w) \right\|_2^2 \\
&= \exp(2\ell(w)) \left( \left\| \sum_k \beta_{k,y} \nabla_w h_k(w) \right\| \right)^2 \\
&\leq \exp(2\ell(w)) \left( \sum_k \beta_{k,y} \| \nabla_w h_k(w) \| \right)^2 \\
&= \exp(2\ell(w)) \left( \sum_k \beta_{k,y} h_k(w) \left\| \tilde{e}^{(k)} - \tilde{h}(w) \right\|_2 \right)^2 \\
&\leq \exp(2\ell(w)) \left( 2 \sum_k \beta_{k,y} h_k(w) \right)^2 \\
&= 4\exp(2\ell(w)) \left( \sum_k \beta_{k,y} h_k(w) \right)^2 \\
&= 4.
\end{aligned}
$$

The first equality follows since the outer product $aa^T$ has one eigenvalue given by $\|a\|_2^2$. The first and second inequalities make use of the triangle inequality. The final equality follows from the definition of $\ell(w)$. Finally, summing the two bounds for the maximum eigenvalues for each component of the Hessian provides the result. □

**Proof of Lemma 10 ($B_H$ bound for deterministic $f|w$):** We prove the lemma for the case of the smoothed log loss, with the case of log loss handled similarly. Let $\tilde{\ell}(w,(x,y)) = -\log\big((1-\alpha)p(y|f(w))+\alpha\big) = const - \log\big(p(y|f(w))+\alpha'\big)$ where $\alpha' = \frac{\alpha}{1-\alpha}$ and the dependence of $f$ on $x$ is suppressed for clarity. Then, $\nabla_w\tilde{\ell}(w) = \frac{\partial\left[-\log(p(y|f)+\alpha')\right]}{\partial f}\nabla_w f(w)$, and

$$\nabla_w^2\tilde{\ell}(w) = \frac{\partial^2\left[-\log(p(y|f)+\alpha')\right]}{\partial f^2}\left(\nabla_w f(w)\right)\left(\nabla_w f(w)\right)^T + \frac{\partial\left[-\log(p(y|f)+\alpha')\right]}{\partial f}\nabla_w^2 f(w). \tag{18}$$

We use the identity $\lambda_{max}(A+B) \leq \lambda_{max}(A) + \lambda_{max}(B)$ to bound each part separately. For the first term $\nabla_w\nabla_w^T$ is positive and therefore $\lambda_{max} \leq c_2 c_1^f$. In the second term, $\nabla_w^2$ is not guaranteed to be positive. We therefore use bounds on the absolute values of the univariate derivative and the eigenvalues (i.e., we use the singular values of the Hessian) to get $\lambda_{max} \leq c_1 c_2^f$. $\square$

**GLM instances:** Letting $\alpha' = \frac{\alpha}{1-\alpha}$, the second derivative of the smoothed log loss is given by

$$\frac{\partial^2\left[-\log(p(y|f)+\alpha')\right]}{\partial f^2} = \frac{1}{(p(y|f)+\alpha')^2}\left(\frac{\partial p(y|f)}{\partial f}\right)^2 - \frac{1}{p(y|f)+\alpha'}\frac{\partial^2 p(y|f)}{\partial f^2}. \tag{19}$$

**Logistic** We use $p(y|f) = \sigma(yf)$ where $y \in \{-1,1\}$ and $\sigma()$ is the sigmoid function. Evaluating the likelihood derivatives, we have

$$\frac{\partial p(y|f)}{\partial f} = y\sigma(yf)(1-\sigma(yf)) \leq \frac{1}{4}, \tag{20}$$

$$\frac{\partial^2 p(y|f)}{\partial f^2} = y^2\sigma(yf)(1-\sigma(yf))(1-2\sigma(yf)). \tag{21}$$

The minimum value of the second derivative (21) is found by optimizing. Letting $\sigma \triangleq \sigma(yf)$, the 3rd derivative of $\sigma$ w.r.t. $f$ is given by

$$\frac{\partial}{\partial f}y^2(\sigma(1-\sigma)(1-2\sigma)) = \frac{\partial}{\partial f}y^2(\sigma - 3\sigma^2 + 2\sigma^3))$$
$$= y^3(\sigma(1-\sigma) - 6\sigma^2(1-\sigma) + 6\sigma^3(1-\sigma))$$
$$= y^3\sigma(1-\sigma)(1-6\sigma+6\sigma^2). \tag{22}$$

Since $\sigma$ is an invertible function of $f$, we can determine the maximum of the second derivative in terms of $\sigma$. The roots of (22) are $0, 1, \frac{1}{2} \pm \frac{\sqrt{3}}{6}$. Plugging the roots back into the second derivative, the minimum value is attained at either $\frac{1}{2} + \frac{\sqrt{3}}{6}$ or $\frac{1}{2} - \frac{\sqrt{3}}{6}$ (depending on the sign of $y$). However, the minimum value always equals $-\frac{\sqrt{3}}{18}$. Therefore, $\frac{\partial^2\left[-\log(p(y|f)+\alpha')\right]}{\partial f^2} \leq c = \frac{1}{16}\frac{1}{(\alpha')^2} + \frac{\sqrt{3}}{18}\frac{1}{\alpha'}$.

**Gaussian** Here, $p(y|f) = \frac{1}{\sqrt{2\pi}\sigma_Y}\exp(-\frac{1}{2}\frac{(y-f)^2}{\sigma_Y^2})$. The likelihood derivatives are given by

$$\frac{\partial p(y|f)}{\partial f} = \frac{1}{\sqrt{2\pi}\sigma_Y^3}\exp(-\frac{1}{2}\frac{(y-f)^2}{\sigma_Y^2})(y-f), \tag{23}$$

$$\frac{\partial^2 p(y|f)}{\partial f^2} = \frac{1}{\sqrt{2\pi}\sigma_Y^3}\exp(-\frac{1}{2}\frac{(y-f)^2}{\sigma_Y^2})\left(-1 + \frac{(y-f)^2}{\sigma_Y^2}\right). \tag{24}$$

From (24), it can be seen that (23) has stationary points at $y - f = \pm\sigma_Y$. The maximum occurs at $y - f = \sigma_Y$, so, $\frac{\partial p(y|f)}{\partial f} \leq \frac{1}{\sqrt{2\pi}\sigma_Y^2}e^{-\frac{1}{2}}$. Also, (24) is bounded from below when $y - f = 0$ which yields a lower bound $-\frac{1}{\sqrt{2\pi}\sigma_Y^3}$. Therefore, $\frac{\partial^2\left[-\log(p(y|f)+\alpha')\right]}{\partial f^2} \leq c = \left(\frac{1}{2\pi\sigma_Y^4 e}\frac{1}{(\alpha')^2} + \frac{1}{\sqrt{2\pi}\sigma_Y^3}\frac{1}{\alpha'}\right)$.

$\square$

**Proof of Corollary 13 (Binary classification in GLM with convex loss):** The loss function $\ell'(w,(x,y)) = \frac{1}{8}(y - (2p(y|w,x)-1))^2$ is bounded between 0 and $\frac{1}{2}$. So, $\frac{1}{\lambda}\Psi \leq \frac{1}{\lambda}\frac{\lambda^2(\frac{1}{2})^2}{2n} = \frac{\lambda}{8n}$.

To find a bound for $B_H$, we derive the Hessian $\nabla_w^2 \ell'(w, (x, y))$ which equals $\frac{\partial^2 \ell'}{\partial p^2}(\nabla_w p(y|w, x))(\nabla_w p(y|w, x))^T + \frac{\partial \ell'}{\partial p}\nabla_w^2 p(y|w, x)$. The 1st and 2nd derivatives of $\ell'$ w.r.t. $p$ are $\frac{\partial \ell'}{\partial p} = \frac{1}{2}(2p - y - 1)$ and $\frac{\partial^2 \ell'}{\partial p^2} = 1$. Letting $\sigma \triangleq \sigma(yw^T x)$, the gradient and Hessian of $p$ w.r.t. $w$ are given by $(1 - \sigma)\sigma y x$ and $(1 - 2\sigma)(1 - \sigma)\sigma y^2 xx^T$. Thus, $\nabla_w^2 \ell'(w, (x, y)) = \left[(1-\sigma)\sigma\right]^2 y^2 xx^T + \frac{1}{2}(2p - y - 1)(1 - 2\sigma)(1 - \sigma)\sigma y^2 xx^T = (1 - \sigma)\sigma y^2 \left[(1 - \sigma)\sigma + \frac{1}{2}(2p - y - 1)(1 - 2\sigma)\right] xx^T \preceq \frac{1}{4}(\frac{1}{4} + 1)xx^T = \frac{5}{16}xx^T$. $\qquad\square$

**Proof of Theorem 14 (Surrogate loss):** Written in terms of $z = \left(\begin{smallmatrix} m \\ vec(V) \end{smallmatrix}\right)$, the loss function evaluates to

$$\frac{1}{2}\log(2\pi) + \frac{1}{2}\log\left(\sigma^2 + \sigma_Y^2 + (a \otimes a)^T P_V z\right) + \frac{1}{2}\frac{(y - a^T P_m z - b)^2}{\sigma^2 + \sigma_Y^2 + (a \otimes a)^T P_V z}, \quad (25)$$

where $P_m$ and $P_V$ are matrices s.t. $P_m z = m$ and $P_V z = vec(V)$. The loss (25) is not convex because of the logarithm in the second term. The surrogate loss $\ell^{surr}()$ (which drops the log) upper bounds (25) and is convex in $z$ since $\frac{\beta(z)^2}{\alpha(z)}$ is convex in $\left(\begin{smallmatrix} \beta(z) \\ \alpha(z) \end{smallmatrix}\right)$ (see e.g., [5]) and $\beta$ and $\alpha$ are each linear in $z$. The derivatives of the convex surrogate w.r.t. $m$ and $vec(V)$ are given by are given by

$$\frac{\partial}{\partial(P_m z)} : -\frac{y - a^T P_m z - b}{\sigma^2 + \sigma_Y^2 + (a \otimes a)^T P_V z}a, \quad (26)$$

$$\frac{\partial}{\partial(P_V z)} : \frac{1}{2}\left(\frac{1}{\sigma^2 + \sigma_Y^2 + (a \otimes a)^T P_V z} - \frac{(y - a^T P_m z - b)^2}{(\sigma^2 + \sigma_Y^2 + (a \otimes a)^T P_V z)^2}\right)(a \otimes a). \quad (27)$$

The Lipschitz bound w.r.t. norm $\|\cdot\|$ for a convex function is equal to the maximum value of the dual norm $\|\cdot\|_*$ of its derivative [19]. Since vector 2-norm is self dual, the Lipschitz bound w.r.t. $z$ is upper-bounded by $\sqrt{\rho_m^2 + \rho_V^2}$ where $\rho_m$ and $\rho_V$ are bounds on the 2-norms of (26) and (27) given by $\rho_m = \frac{1}{\sigma_Y^2}\max_{x \in X}\|a\|_2 \max_{x \in X, y \in Y, m}|m^T a + b - y|$, and $\rho_V = \frac{1}{2\sigma_Y^2}\max_{x \in X, y \in Y, m}\|a\|_2^2\left(1 + \frac{(y - a^T m - b)^2}{\sigma_Y^2}\right)$. Hence, the convex surrogate is $\rho$-Lipschitz w.r.t. 2-norm in $\left(\begin{smallmatrix} m \\ vec(V) \end{smallmatrix}\right)$ with $\rho = \sqrt{\rho_m^2 + \rho_V^2}$.

Noting that the regularizer is $\frac{1}{2}$-strongly convex, we can apply Theorem 1 to get

$$\mathop{\mathrm{E}}_{S \sim D^n}[r_{2Ds}(q_{2Ds}^\star)] \leq \mathop{\mathrm{E}}_{S \sim D^n}[r_{2Bay}^{surr}(q_{2Ds}^\star)] \leq \min_{q \in Q} r_{2Bay}^{surr}(q(w)) + \frac{1}{\eta n}\left(B_m^2 + B_V^2\right) + 8(\rho_m^2 + \rho_V^2)\eta. \quad (28)$$

Setting $\eta = \frac{1}{\sqrt{n}}$ and utilizing Markov's inequality yields the result. $\qquad\square$

## C  Details for Section 5

The results in this paper expose the fact that different variational algorithms are apparently implicitly optimizing criteria for different loss functions. In particular, $q_{2A}^\star$ optimizes for $r_{2A}$, $q_{2Bi}^\star$ optimizes for $r_{2Gib}$ and $q_{2D}^\star$ optimizes for $r_{2Bay}$. Even though we were able to bound $r_{2Bay}$ of the $q_{2Bi}^\star$ algorithm, it is interesting to check the performance of these algorithms in practice.

We present an experimental study comparing these algorithms on the correlated topic model (CTM).

$$p(w) = \mathcal{N}(w|\mu, \Sigma), \quad p(f|w) = \prod_i \mathrm{Discrete}(f_i|h(w)), \quad p(y_i|f_i) = \mathrm{Discrete}(y_i|\beta_{f_i, y_i}), \quad (29)$$

where $h()$ is the logistic transformation.

For this model the simple loss $\ell_{2A}$ is

$$\ell_{2A}(q, y_i) = \int\left(\sum_k -\log\left(\beta_{k, y_i}\right)h_k(w)\right)q(w)dw, \quad (30)$$

| "Posterior" | $\sum_{y_i \in \text{test}} \ell_{2\text{A}}(y_i)$ | $\sum_{y_i \in \text{test}} \ell_{2\text{Gib}}(y_i)$ | $\sum_{y_i \in \text{test}} \ell_{2\text{Bay}}$ |
|---|---|---|---|
| $q_{2\text{A}}^\star$ | 2960.624 (0.119) | 2956.082 (0.159) | 2955.171 (0.321) |
| $q_{2\text{Bi}}^\star$ | 3058.442 (2.846) | 2920.572 (0.581) | 2915.615 (0.578) |
| $q_{2\text{D}}^\star$ | 3066.244 (6.034) | 2931.522 (1.339) | 2915.317 (0.580) |

Table 1: *Artificial data*. Final (converged) cumulative loss of model learned by different variational algorithms. The cumulative loss is evaluated on **test**. The values represent the mean of 30 trials (1-$\sigma$ values in parentheses). Cumulative $\ell_{2\text{Bay}}$ of the generative model evaluated on test is 2912.304.

For the other two losses, note that we can explicitly integrate out $f$ to get a closed form expression for $p(y_i|w) = \sum_k \beta_{k,y_i} h_k(w)$. This yields

$$\ell_{2\text{Gib}}(q, y_i) = \int - \log \left( \sum_k \beta_{k,y_i} h_k(w) \right) q(w)dw, \tag{31}$$

$$\ell_{2\text{Bay}}(q, y_i) = - \log \int \left( \sum_k \beta_{k,y_i} h_k(w) \right) q(w)dw. \tag{32}$$

To explore the relation between the algorithms and their performance we run the three algorithms and report their empirical risk on a test set, where the risk is also measured in three different ways. We repeat this on one 1000-word document artificially created with the generative model and one real 1300-word document randomly selected from the *nips* dataset [13].

Each document is randomly split into equal size train and test sets. Optimizing the three objectives on the training set results in three posteriors which are then evaluated on the test set. We use $K = 50$ topics and the hyperparameters $\mu, \Sigma, \beta$ are fixed to the parameters of the generative model in the artificial data, and to values inferred from a larger subset of the *nips* dataset (excluding the previously selected test document).

The optimization algorithm is SGD in the mean $m$ and Cholesky factor $C$ of the covariance $V$. Equation 7 of [18] is used to calculate the derivative w.r.t. the mean, and equation 10 of [18] is used to calculate the derivative w.r.t. the Cholesky factor. We use 10 Monte Carlo samples to compute derivatives during training. The experiment is repeated 30 times and averages and standard deviations across these runs are reported. Test set losses (30-32) are not available in closed form, so we use 1000 Monte Carlo samples to calculate loss function values (the use of $10^4$ samples produced visually indistinguishable results).

Results for the 1000-word artificial document are presented in Table 1 and results for the *nips* document are shown in Table 2. The tables show results at convergence where we can clearly see that indeed each algorithm is best at optimizing its own implicit criterion. However, considering $r_{2\text{Bay}}$, the differences between the outputs of the variational algorithm $q_{2\text{Bi}}^\star$ and direct loss minimization $q_{2\text{D}}^\star$ are relatively small. Figure 1 shows the corresponding training curves on the artificial document. We see that at least in this case $q_{2\text{Bi}}^\star$ takes longer to reach the optimal point for $r_{2\text{Bay}}$. Clearly, except for its own implicit criterion, $q_{2\text{A}}^\star$ should not be used. This agrees with prior empirical work on $q_{2\text{A}}^\star$ and $q_{2\text{Bi}}^\star$ [22]. The current experiment shows the potential of direct loss optimization for improved performance but justifies the use of $q_{2\text{Bi}}^\star$ both under correct model specification (artificial data) and when the model is incorrect (real data).

| "Posterior" | $\sum_{y_i \in \text{test}} \ell_{2\text{A}}(y_i)$ | $\sum_{y_i \in \text{test}} \ell_{2\text{Gib}}(y_i)$ | $\sum_{y_i \in \text{test}} \ell_{2\text{Bay}}$ |
|---|---|---|---|
| $q_{2\text{A}}^{\star}$ | 5447.317 (0.128) | 5423.254 (0.855) | 5418.343 (1.403) |
| $q_{2\text{Bi}}^{\star}$ | 5708.557 (3.625) | 5029.380 (4.998) | 5020.091 (2.248) |
| $q_{2\text{D}}^{\star}$ | 5719.236 (5.433) | 5062.755 (3.065) | 5016.625 (1.077) |

Table 2: *Real data*. Final (converged) cumulative loss of model learned by different variational algorithms on one document from *nips* dataset. The cumulative loss is evaluated on **test**. The values represent the mean of 30 trials (1-$\sigma$ values in parentheses).

## Footnotes

[4]The interchange of the expectations over $p(w)$ and $D^n$ is justified by a special case of Fubini's theorem for non-negative functions.