[Reviews · NeurIPS 2017]

Reviewer 1



This work proposes a risk bound for various variational bayes algorithms by leveraging the work of [5, 14, 9]. Those bounds show that the variational bayes posterior is competitive with any point estimate of the same class. In section 4, the author specialize the bound to variational algorithms which have a Normal Distribution as the prior and posterior. To avoid the max_q term diverging to infinity, the posterior family is limited to covariance matrices with their smallest eigenvalue above a threshold. At line 230, the bounded loss is considered and a quick analysis is made. However, having an explicit writeup of the bound would be useful. Also, calculating real values of the bound in the proposed experiment would be useful. The main result is not trivial and somewhat useful in a theoretical sense. However, the main result is buried behind maths. Stating more clearly the final bound and resolving the \psy function and \lambda to their actual values according to the choice of either bounded loss or sub-gaussian would be important. Also discussing the fact that bounded loss is unrealistic in the case of NLL and that sub-gaussian bounds don’t converge to 0 as the number of samples increase as stated in [5]. I would also like the author to discuss more in details the usefulness of this result. Minors: * Line 73: p(y|w,x) instead of p(y|w) ? * Between 120 and 130 there are a few places where it feels like it should be conditioned on x but it’s not. Can we be more consistent?

Reviewer 2



General comments: Overall this is an interesting paper with some nice theoretical results, though the techniques are rather standard. As far as I know, the results on risk bounds for Bayes predictor are novel. It provides theoretical evidence for Bayesian treating of parameters (at least for Gaussian priors) and the generalization guarantee of variational approximations (1-layer, also 2-layer by collapsing the second), which should be of wide interest to the Bayesian community. The experiments on CTMs demonstrate the risk bounds of different variational approximations, though I find the direct loss minimization case less significant because usually it’s hard to compute the marginal log likelihoods. The main concern that I have is the presentation of the results. It seems that the authors induce unnecessary complexity during the writing. The three kinds of risks induced in the paper ($r_{Bay}, r_{Gib}, r_{2A}$) are just negative expectations of marginal log likelihood $\log p(y|x)$, the first term in elbo for w, and the first term in elbo for w and f (using terminology from variational inference). In Section 2.2, a 2-layer Bayesian model is introduced. I understand the authors’ intention to align with real model examples used in the paper after I finish later sections. But I still feel it’s better to derive the main results (Th.2, 6, 8) first on the 1-layer model $p(w)p(y|x,w)$ and later extend them into 2-layer settings. In fact the main theoretical result is on collapsed variational approximations, which can be seen as a 1-layer model (This is also how the proof of Corollary 3 comes). Minor: In Section 2.2, if the 2-layer model is defined as $p(w)p(f)\prod_i p(y_i|f_i, x_i)$, should the risks be $...E_{p(f|w)} p(y|f, x)$ instead of $...E_{p(f|w,x)}p(y|f)$? I’m not sure why there is a dependence of f on x.

Reviewer 3



This paper considers several versions of variational inference in one-level and two-level latent variable models: pointing out the loss functions they implicitly minimize, proving risk bounds, and testing their performance empirically. The exposition is clear and the contributions seem significant, though I do not have the background knowledge to evaluate them in more detail. Typographical comments: l53: define -> defines l58: yet -> yet have l127: comma belongs on previous line l134: desribed -> described l145: remove colon after footnote l164: comma missing at end of line References: please add {}'s to ensure capitalization of PAC, Bayes, Gibbs, ...